# Top-down and bottom-up interactions rely on nested brain oscillations to shape rhythmic visual attention sampling

Jelena Trajkovic[1]*, Domenica Veniero[2], Simon Hanslmayr[3], Satu Palva[3,4], Gabriela Cruz[3], Vincenzo Romei[5,6], Gregor Thut[3,7]*

1 Department of Cognitive Neuroscience, Faculty of Psychology and Neuroscience, Maastricht University, Maastricht, Netherlands, 2 School of Psychology, University of Nottingham, Nottingham, United Kingdom, 3 Centre for Cognitive Neuroimaging, School of Psychology and Neuroscience, University of Glasgow, Glasgow, United Kingdom, 4 Neuroscience Center, Helsinki Institute of Life Science, University of Helsinki, Helsinki, Finland, 5 Dipartimento di Psicologia, Centro studi e ricerche in Neuroscienze Cognitive, Alma Mater Studiorum – Università di Bologna, Campus di Cesena, Cesena, Italy, 6 Facultad de Lenguas y Educación, Universidad Antonio de Nebrija, Madrid, Spain, 7 Centre de Recherche Cerveau et Cognition (CerCo), CNRS UMR5549 and Université de Toulouse, Toulouse, France

* gregor.thut@glasgow.ac.uk (GT); jelena.trajkovic@maastrichtuniversity.nl (JT)

## Abstract

Adaptive visual processing is enabled through the dynamic interplay between top-down and bottom-up (feedback/feedforward) information exchange, presumably propagated through brain oscillations. Here, we causally tested for the oscillatory mechanisms governing this interaction in the human visual system. Using concurrent transcranial magnetic stimulation-electroencephalography (TMS-EEG), we emulated top-down signals by a single TMS pulse over the frontal eye field (right FEF), while manipulating the strength of sensory input through the presentation of moving concentric gratings (compared to a control-TMS site). FEF-TMS without sensory input led to a top-down modulated occipital phase realignment, alongside higher fronto-occipital phase connectivity, in the alpha/beta band. Sensory input in the absence of FEF-TMS increased occipital gamma activity. Crucially, testing the interaction between top-down and bottom-up processes (FEF-TMS during sensory input) revealed an increased nesting of the bottom-up gamma activity in the alpha/beta-band cycles. This establishes a causal link between phase-to-power coupling and top-down modulation of feedforward signals, providing novel mechanistic insights into how attention interacts with sensory input at the neural level, shaping rhythmic sampling.

## Introduction

Flexible and adaptive perception is made possible by top-down (e.g., predictive) regulation of sensory input. This involves a large-scale brain network, in which top-down

**Data availability statement:** All data reported in this paper have been made publicly available through the Open Science Framework: https://osf.io/9uhta/. Source code is available through Zenodo: 10.5281/zenodo.14836618.

**Funding:** This work was supported by the Medical Research Council to GT and SP (MR/V003623/1, https://www.ukri.org/councils/mrc/). V.R. is supported by MUR – Ministry of University and Research, Italy (P2022XAKXL and 2022H4ZRSN, https://researchitaly.mur.gov.it/en/homepage-en/) and by the Ministerio de Ciencia, Innovación y Universidades, Spain (PID2019-111335GA-100, https://www.ciencia.gob.es/en/). The funders had no role in study design, data collection and analysis, decision to publish, or preparation of the manuscript.

**Competing interests:** I have read the journal's policy and the authors of this manuscript have the following competing interests: SH is an Editorial Board Member of PLOS Biology.

**Abbreviations :** CBA, cortex-based alignment; EEG, electroencephalography; FEF, frontal eye field; ICA, individual component analysis; ITPC, intertrial phase coherence; MI, modulation index; MSO, maximal stimulator output; PAC, phase-amplitude coupling; TESA, TMS-EEG signal analyzer; TF, time-frequency; TMS, transcranial magnetic stimulation; V5, visual cortex; wPLI, weighted phase lag index.

signals influence lower-level visual areas [1–4]. For example, when attending to a specific spatial location, top-down attentional control enables enhanced processing of information in the attentional spotlight [5]; with the frontal eye field (FEF) being a key area for this control [6]. Specifically, non-human primate work has demonstrated that task-related attentional signals generated in the FEF exert top-down influence on visual areas [7–9], and that micro-stimulation of FEF yields attention-like effects on the visual system [10,11]. Similarly, a causal role of the FEF in attentional control has been demonstrated in human participants, in whom the activation of FEF by means of non-invasive transcranial magnetic stimulation (TMS) causes changes in visual cortex activity [12–14] and perception [15–17].

At the same time, brain oscillations have been identified to play a role in attention and perception processes, whereby attention-controlled oscillatory phase synchronization in specific frequency bands both within and between brain regions has been shown to facilitate visual processing [18–20]. For instance, non-human primate studies found that tasks emphasizing top-down processing evoke stronger synchronization at lower frequencies (alpha/beta), while feedforward processing is associated with high-frequency (gamma) oscillations [7,21,22]. In support of this dichotomy, micro-stimulation of pathways and cortical layers involved in feedback or feedforward signaling is leading to stronger synchronization at alpha/beta or gamma frequencies, respectively [22–26]. Similarly, in the human visual system, inter-areal rhythmic influences are observed to predominate in the alpha/beta bands along feedback projections [18,27,28], but in the gamma band along feedforward projections [19,28]. This accords with earlier findings revealing that sensory processing in primary visual cortices is related to gamma activity [29–32], with this activity being modulated by both low-level physical features (such as stimulus size and spatial location [33]) and attentional demands [34,35]. Finally, causal probes of top-down signaling through FEF stimulation by TMS in human participants – mirroring previous micro-stimulation probes in animals [10,23] – point to top-down control being enabled by frequency-specific oscillatory changes between distant brain areas at alpha/beta frequency, namely their phase realignment [36].

Taken together, the above findings suggest that top-down control from higher-order areas is implemented through oscillatory processes at alpha/beta frequency, while sensory input is processed and forwarded through fast oscillatory gamma activity. However, how the top-down rhythms interact with bottom-up brain signals is still unknown. One candidate mechanism is cross-frequency interactions, where the slower (alpha/beta frequency) oscillations organize gamma activity through phase-amplitude coupling (PAC) such that gamma bursts become nested within the alpha/beta phase, thereby enabling a coordinated interplay between neural elements of a large-scale attention network [37,38]. By extension, PAC of alpha/beta to gamma oscillations could enable a coupling between feedback- and feedforward streams for optimal performance. Yet, causal tests of such a mechanism of top-down control over bottom-up input are lacking.

The aim of this study was to causally probe this interaction in the human brain using concurrent TMS-EEG. Specifically, we stimulated FEF through a TMS pulse

causing top-down influences in the alpha/beta range over lower-level visual areas (replicating [36]), while manipulating sensory input in the form of a continuously presented moving sine grating. By simultaneously recording electroencephalography (EEG), we tested whether the FEF stimulation reorganized the sensory-driven gamma activity to become phase-coupled to the top-down generated alpha/beta-cycles as an expression of top-down interactions over bottom-up signals.

## Results

In healthy human volunteers (*N* = 30), we employed brief TMS pulses to activate an attention control center in the prefrontal cortex (FEF-TMS) and compared its effects to a control stimulation site (M1foot-TMS). At the same time, we manipulated the strength of sensory input through the presentation of moving concentric gratings (GRATING± condition) that lasted for 5 s. The participants were instructed to respond with a button press every time they perceived a pause (glitch) in the motion of the grating (see Fig 1). The duration of the glitch was thresholded for each participant (for more details, see Methods). We then tracked the neural expressions of the associated top-down and bottom-up signals with concurrent EEG.

### Sensory input induces gamma-band activity

Our first analysis aimed at confirming that our experimental manipulation of sensory input had the expected consequences, i.e., that the moving grating enhanced gamma activity over posterior sites. To do so, all trials with versus without the moving grating (GRATING±) were compared in a period before any TMS pulse and the occurrence of possible motion glitches (first second after moving grating onset). The permutation-based analysis confirmed that there is significantly higher gamma synchronization in trials with the grating stimulus present as compared to the no-grating condition, lasting

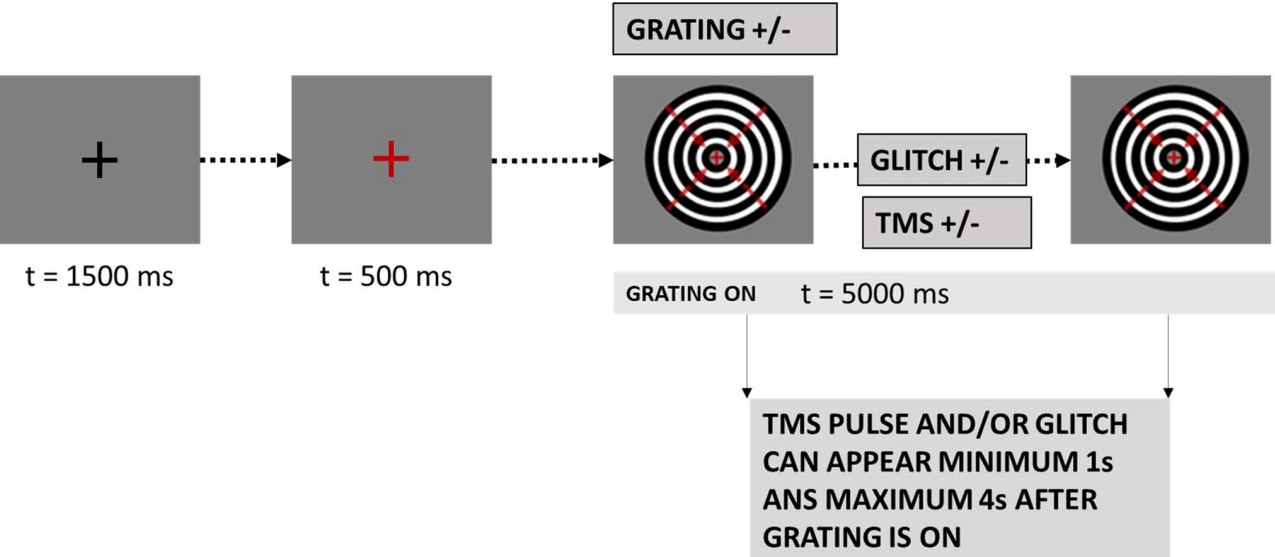

**Fig 1. Experimental and task design.** Trial sequence. Each trial began with a black fixation cross lasting 1,500 ms. Afterward, the cross turned red indicating the beginning of the trial. After 500 ms, the visual grating stimulus was presented. It consisted of a black and white sine grating contracting inward and was displayed for 5 s. In the grating-absent condition, the red fixation cross was displayed for the same amount of time. Some grating-present trials were followed by a target glitch in the grating motion. Participants had to respond with a button press whenever they perceived the glitch. There were three TMS conditions: right FEF-TMS, control-TMS over the right foot area, and no TMS. Both the TMS pulse and the target glitch could appear from 1 s earliest to 4 s latest after grating onset.

PLOS Biology

for the whole post-grating time window analyzed (0–800 ms) (Fig 2A, see also S1 Fig for the same analysis on the whole grating duration: 0–5,000 ms). These differences span over both low and higher gamma frequency bands (50–80 Hz). Importantly, differences in map topography indicate that these effects were constrained to posterior electrodes (see Fig 2B). As expected, grating presentation was also accompanied by a desynchronization across lower 10–20 Hz (alpha-beta) frequencies.

### FEF exerts top-down control over posterior areas through phase realignment and enhanced connectivity in alpha/beta frequency bands

We investigated the influence of FEF-TMS on posterior activity via the analysis of inter-trial phase coherence (ITPC), in comparison to a control-TMS site (TMS of nearby M1-foot area). Based on a previous report [36], we expected FEF activation by TMS to lead to a posterior phase reset of oscillatory activity in the alpha/beta frequency band. Here, we replicate these findings. In trials without a grating stimulus (GRATING−), we observed a significantly higher ITPC for FEF-TMS versus control-TMS in the lower beta frequency band (13–20 Hz) over posterior sites in the right (stimulated) hemisphere, which lasted until around 400 ms after the TMS pulse (Fig 3A). In analogy, for trials with a grating stimulus present (GRATING+), FEF-TMS, relative to control-TMS, led to higher ITPC over right posterior sites in a similar lower beta frequency range, including also higher alpha-band activity (11–19 Hz), and this for mainly the first 250 ms after the TMS pulse (Fig 3B). There were no significant effects across the left posterior electrodes, in either grating-present or grating-absent condition.

## TIME-FREQUENCY AMPLITUDE

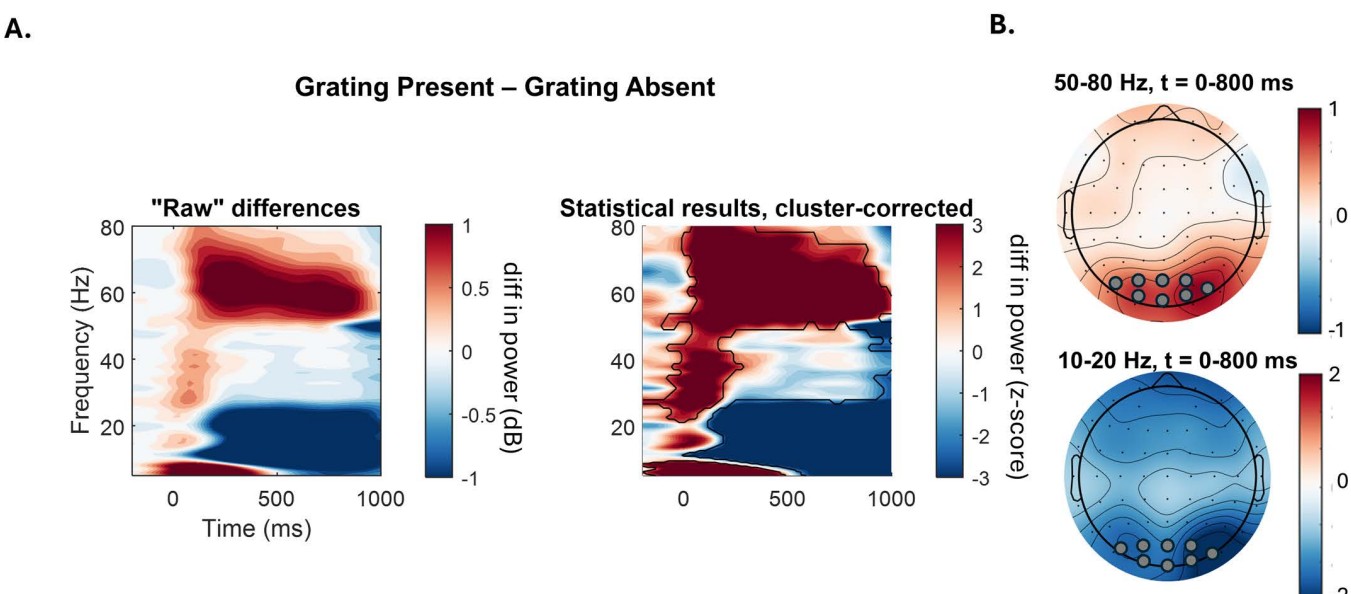

**Fig 2. Time-frequency analysis: Grating effects. (A)** Left panel: Power differences in time-frequency space over posterior electrodes (O2, O1, POz, Oz, PO8, PO7, PO4, PO3) between the grating-present and grating-absent condition (GRATING±). The frequency for the analysis (y-axis) ranged from 5–80 Hz, while the time (x-axis) ranged from −200 to 800 ms, where 0 is the time point of grating onset. Right panel: Z-scores of the permutation-based analysis between grating-present and grating-absent condition. Significant clusters are framed with the black line. **(B)** Topographies of the significant clusters with amplitude differences in the gamma (upper map) and higher alpha/lower beta frequency bands (lower map). Data underlying this figure can be found at https://osf.io/9uhta/. Electrodes used for the time-frequency analysis are marked with gray circles. diff = difference; dB = decibel; Hz = hertz; *t* = time.

## INTER-TRIAL PHASE COHERENCE (ITPC)

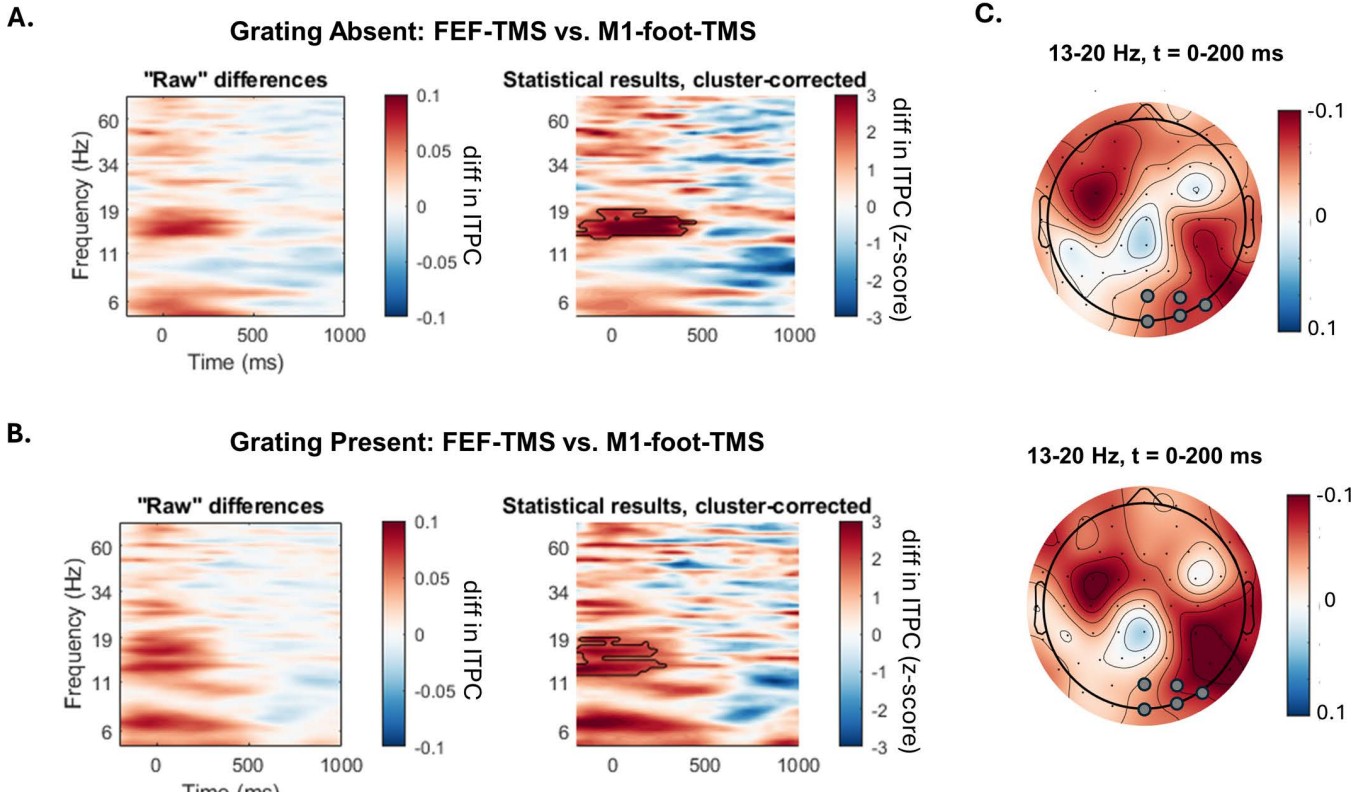

**Fig 3. Inter-trial phase coherence (ITPC) analysis: Differences between FEF-TMS and control-TMS (over M1foot). (A)** Grating-absent condition. Left panel: ITPC differences of the posterior cluster in the stimulated (right) hemisphere (electrodes: O2, POz, Oz, PO8, PO4) between FEF-TMS and control-TMS. The time range for the analysis (x-axis) is from −200 ms to 1,000 ms, where 0 is the time point of the TMS pulse. The frequency (y-axis) ranged from 5–80 Hz. Right panel: Z-scores of the permutation-based statistical analysis between FEF-TMS and control stimulation (M1foot-TMS). Significant clusters are framed with black lines. **(B)** Grating-present condition. Left panel: Raw differences in ITPC plots of the posterior cluster in the stimulated (right) hemisphere (electrodes: O2, POz, Oz, PO8, PO4) between FEF-TMS and control stimulation (M1foot-TMS). The frequency for the analysis (y-axis) ranged from 5–80 Hz. The time range (x-axis) is from −200 ms to 1,000 ms, where 0 is the time point of the TMS pulse. Right panel: Z-scores of the permutation-based analysis between FEF-TMS and control stimulation (M1foot-TMS). **(C)** Topographies of the significant clusters of the identified ITPC differences in the grating-absent (upper map) and grating-present condition (lower map). Data underlying this figure can be found at https://osf.io/9uhta/. Electrodes used for the ITPC analysis are marked with gray circles. diff = difference; Hz = hertz; *t* = time.

To assess if FEF-TMS (versus foot-TMS) enhances inter-areal phase coupling between frontal and posterior sites, we calculated the weighted phase lag index (wPLI): a phase lag-based measure not affected by volume conductance. Subsequently, non-parametric permutation analysis was used to compare wPLI values across the two stimulation sites (FEF-TMS, M1-foot) in the two grating conditions (GRATING+ and GRATING−). Connectivity was estimated in the 300 ms window following TMS, between frontal and parieto-occipital electrodes, both in the TMS-stimulated and non-stimulated hemisphere, in the frequency band identified as significant via ITPC. The results indicate that there is enhanced fronto-posterior connectivity for FEF-TMS as compared to control-TMS. Importantly, these differences were only significant for fronto-posterior electrode pairs in the right (stimulated) hemisphere (Fig 4A). As expected, these differences were visible both in the grating stimulus present (Fig 4C) and absent (Fig 4B) condition, as they likely represent a top-down control mechanism that should be present independently of sensory input. However, we note that the patterns of connectivity effects vary between the GRATING+ and GRATING− conditions (Fig 4B versus Fig 4C). This is to be expected given

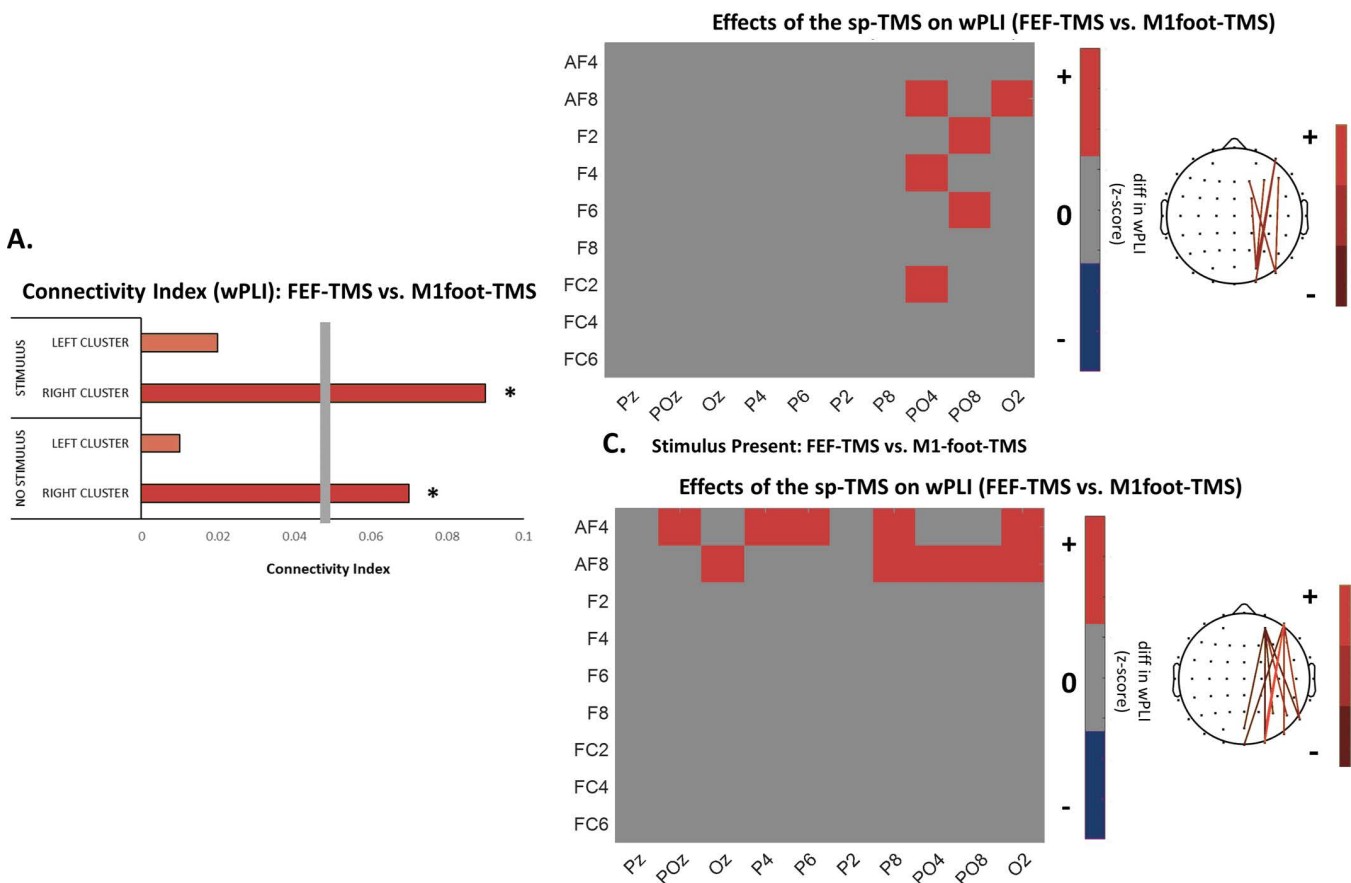

**Fig 4. Changes in interregional coupling between fronto-posterior electrode pairs when contrasting activity recorded during FEF-TMS and control-TMS (over M1foot). (A)** Connectivity index for each of the experimental conditions (grating-present vs. absent) and the two hemispheres (right, stimulated and the left, non-stimulated hemisphere); gray vertical bar shows the statistical threshold. Red ink indicates increases in interregional coupling for the FEF-TMS vs. control stimulation. **(B)** Grating-absent condition. Left: Connectivity matrix measured by the weighted phase lag index (wPLI) across all electrodes included in the regions of interest for the grating-absent condition in the higher alpha/lower beta range (13–20 Hz) and the right (stimulated) hemisphere. Red ink indicates significant increases in interregional coupling for the FEF-TMS vs. control stimulation. Right: Topographical representations of the electrode pairs showing significantly increased coupling for FEF-TMS vs. control-TMS. **(C)** Grating-present condition. Left: Connectivity matrix (measured by the wPLI) across all electrodes included in the regions of interest for the grating-present condition in higher alpha/lower beta range (13–20 Hz) and the right (stimulated) hemisphere. Red ink indicates significant increases in interregional coupling for the FEF-TMS vs. control stimulation. Right: Topographical representations of the electrode pairs showing significantly increased coupling for FEF-TMS vs. control-TMS. Data underlying this figure can be found at https://osf.io/9uhta/.

that TMS effects are known to depend on the prior state of the brain, as evidenced for instance by brain state-dependent influences on the propagation of TMS-induced local action potentials [39–42] that can be driven by the experimental manipulation of sensory input [12], closely mirroring our findings.

## Top-down control and sensory processing interact through cross-frequency coupling

So far, we have demonstrated gamma-band activity over posterior sites to be related to bottom-up signaling of sensory input and/or sensory processing. Independently, we found that fronto-to-posterior phase realignment and connectivity enhancement – both in the alpha/beta band – are related to top-down influences. We next tested whether the top-down signal may structure bottom-up input through PAC, potentially enabling the integration of these two mechanisms of

information processing. If so, we would expect gamma activity related to sensory input to be nested in distinct phases of the lower (alpha/beta) frequency, i.e., the bottom-up input to be top-down phase-controlled by FEF-generated signals.

To investigate this hypothesis, we calculated the modulation index (MI) of PAC for the 200 ms following the TMS pulse, between low-frequency and high-frequency activity (frequency ranges tested: 5–20 Hz versus 30–80 Hz), across right parieto-occipital sites. Our results revealed an increase of PAC over posterior electrodes when FEF is activated by TMS (versus control-TMS over the M1-foot area) for the case when the stimulus was present (GRATING+, Fig 5B). More specifically, gamma amplitude (from 60–80 Hz, see the framed significant cluster) was coupled to the lower beta phase (at 13–20 Hz), with a higher gamma activity for the bins around the beta peak (Fig 5D and 5E). Crucially, these differences between the two TMS protocols were not present in the absence of sensory input (GRATING−, Fig 5A), which shows that both top-down and bottom-up signals have to be present for PAC to occur. This suggests that alpha/beta to gamma PAC is indeed a mechanism that allows both processes to interface.

## Discussion

Perception is much more than the passive reception of sensory input; it is the result of the complex interactions of internal signals based on priors with the incoming sensory input [43,44]. Brain oscillations may enable this interplay, where feedback and feedforward processing are carried by slower and faster oscillatory activity, respectively, and where these two streams of processing may interact through cross-frequency mechanisms [22–25,28,45,46]. Yet, a causal test of how feedback routes the sensory-driven signal through oscillatory mechanisms is lacking. Here, we tested whether the generation of a top-down signal causes visually-driven, high-frequency activity in the posterior cortex to be nested to the phase of top-down generated low-frequency oscillations. To this end, we emulated an attentional impulse via FEF stimulation through a TMS pulse in the presence versus absence of sensory activity, while examining effects on EEG.

Our results first confirmed that a continuous visual input, in the form of a moving sine grating [47], results in synchronization of gamma-band activity across the posterior brain areas, corroborating prior evidence that the feedforward drive of sensory input is accompanied by gamma synchronization [29–32]. This gamma increase is thought to enhance synaptic summation [19], thereby boosting the transfer of information at hand [48]. Second, when emulating an attentional impulse by FEF-TMS (versus TMS over a control area), we observed a high alpha/low beta oscillatory phase realignment and enhanced connectivity between the frontal and posterior EEG sensors both when sensory input was present and absent. This is in line with numerous studies indicating that alpha/beta activity reflects top-down processes [7–9] and confirms the notion that reorganization of alpha/beta phase, through long-range connectivity in the same frequency band, may represent a basic mechanism for feedback communication [36]. The anatomical specificity of the effects in posterior and frontal regions is further in line with a recent framework emphasizing that feedforward and feedback interactions are confined to distinct cortical networks [49]. Third, in the same posterior cluster, when FEF-TMS was combined with sensory input, we observed gamma activity to be locked to the phase of the alpha/beta frequency (as compared to TMS over the control site), but not when no sensory input was present. Importantly, the coupled low/high-frequency oscillations exactly matched in frequency the alpha/beta- and gamma-range identified as significantly associated with top-down and bottom-up sensory processing in isolation. This pattern of results therefore establishes a causal link between cross-frequency coupling and top-down modulation of feedforward signals and corroborates that this top-down versus bottom-up integration is achieved through phase-to-amplitude cross-frequency coupling.

The alpha-oscillation (and adjacent brain rhythms) has not only been identified as governing feedback processing streams but has also been shown to be perceptually relevant. More specifically, its phase across visual areas has been related to perceptual outcome, where threshold stimuli are sometimes perceived and sometimes missed based on the instantaneous phase at the moment of stimulus presentation [50–55]. Accordingly, it has been proposed that these fluctuations reflect the cyclic changes in brain excitability indicating the rapid waxing and waning of visual and/or attentional sampling, predicting perceptual outcomes [51,53,56]. While research on visual/attentional sampling has mostly pointed

## PHASE-TO-AMPLITUDE COUPLING (PAC)

**A.** Grating Absent: FEF-TMS vs. M1-foot-TMS

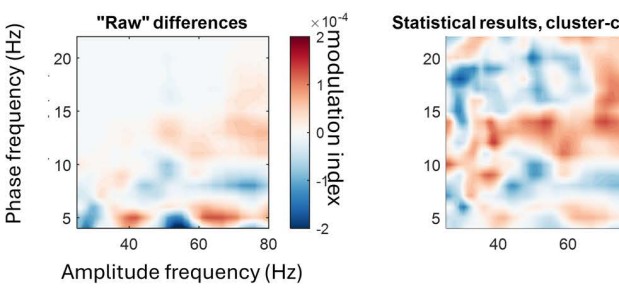

**C.**

Grating Absent: FEF-TMS vs. M1-foot-TMS
Phase-frequency: 13-20 Hz; amplitude-frequency: 60-80 Hz

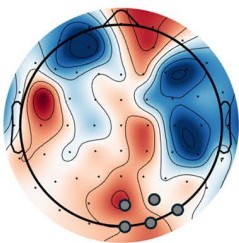

**B.** Grating Present: FEF-TMS vs. M1-foot-TMS

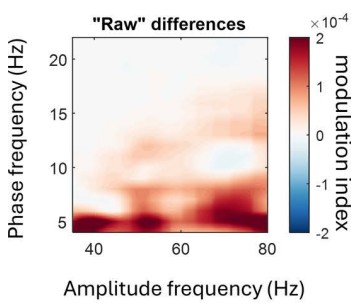

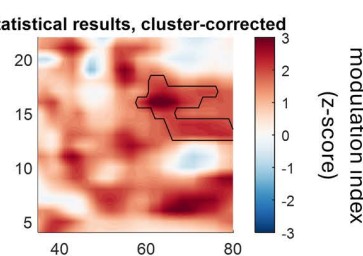

Grating Present: FEF-TMS vs. M1-foot-TMS vs.
Phase-frequency: 13-20 Hz; amplitude-frequency: 60-80 Hz

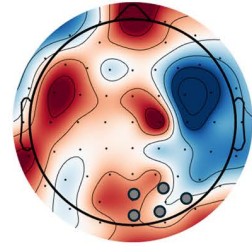

**D.**

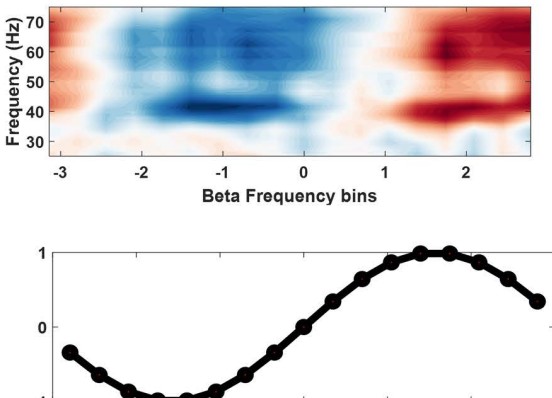

**E.**

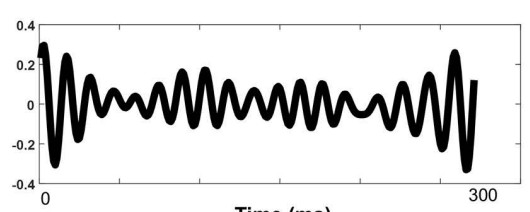

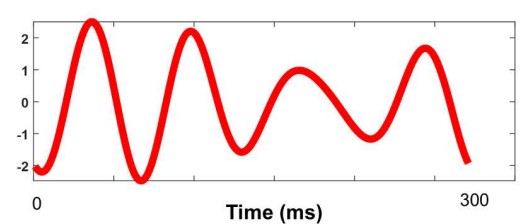

**Fig 5. Phase-to-amplitude coupling (modulation index [MI]) post-TMS pulse (0–300 ms). (A)** Grating-absent condition. Left panel: Raw difference plots of the MI in the posterior cluster over the stimulated (right) hemisphere (electrodes: O2, POz, Oz, PO8, PO4) between FEF-TMS and control stimulation (M1foot-TMS). The frequency of the lower alpha/beta phase is shown in the y-axis. The frequency of the higher gamma amplitude is represented on the x-axis. Right panel: Z-scores of the permutation-based analysis between FEF-TMS and control stimulation (M1foot-TMS). No significant clusters were identified. **(B)** Grating-present condition. Left panel: Raw difference plots of the MI in the posterior cluster of the stimulated (right) hemisphere (electrodes: O2, POz, Oz, PO8, PO4) between FEF-TMS and control stimulation (M1foot-TMS). The frequency of the lower alpha/beta phase is represented on the y-axis. The frequency of the higher gamma amplitude is shown on the x-axis. Right panel: Z-scores of the permutation-based analysis between FEF-TMS and control stimulation (M1foot-TMS). Significant clusters are framed with the black line. **(C)** Topographies of the significant clusters of the identified MI differences in the grating-present vs. grating-absent condition. Electrodes used for the MI analysis are marked with gray circles. **(D)** Beta (15 Hz) phase sorted gamma power is shown for all data of the experimental condition that yielded a significant MI (FEF-TMS when stimulus is present).

Note the higher gamma activity for the bins around the beta peak. **(E)** Data for a single trial from the experimental condition that yielded significant MI (FEF-TMS when grating is present) filtered in the gamma and beta frequency. Note that gamma increases mostly in beta peaks. Data underlying this figure can be found at https://osf.io/9uhta/. Hz = hertz; *t* = time.

to effects in the alpha- or theta-range [57–60], it is conceivable that the phase reorganization in the high alpha/low beta band found here reflects the expression of another rhythmic sampling mechanism at the interface between attention and sensory processing, that comes about through feedback-controlled periodic modulation of feedforward signaling. In support of this view, we recently found that activation of the FEF via a single TMS pulse not only phase-resets posterior alpha/beta-oscillations but induces cyclic modulations of visual perception and extrastriate visual cortex (V5) excitability at high alpha/low beta frequency, as tested by motion discrimination accuracy and moving phosphene perception [36]. Interestingly, it has recently been proposed that attentional sampling is organized via two alternating attentional states [61,62]. While one attentional state is thought to be related to alpha-band activity and the rhythmic (pulsed) suppression of visual processing, the second state, associated with better stimulus detection, is characterized by increased posterior gamma activity, alongside enhanced beta activity in FEF [62]. According to this model, gamma activity enables enhanced sensory processing at the cued location, while beta activity leads to a decreased likelihood of shifting away from the attentional locus [61,62]. In line with this model, our data suggests that beta-gamma interplay is indeed a fingerprint of FEF-related visual attention processes, that seem to shape rhythmic sampling at high alpha/low beta frequency through frequency-specific feedback-over-feedforward interactions. Please note that the current experimental design was optimized for tracking EEG changes in response to FEF activation and sensory input but not the associated behavioral effects, since we sampled behavior only for one-time point, namely for target stimuli coinciding with the TMS pulse. Presenting a visual target serves to ensure that participants remain engaged throughout the experiment but the simultaneity of FEF activation and target presentation prevents TMS-induced activation to interact with target processing. However, given that we here replicate the FEF-TMS-induced phase realignment at high alpha/low beta frequency over occipital EEG-sites observed previously [36], while showing this to interact with visual gamma activity, we would expect perceptual performance to co-fluctuate with these effects, i.e., the behavioral effects not to differ from previous work [36].

The framework positing that the communication between brain regions is based on nested oscillations [37,46] accommodates prominent and related theories of oscillatory communication. The first one is the communication-through-coherence theory, which posits that interregional communication is considered established when brain regions oscillate at the same frequency with a stable phase lag [63]. The second one is the gating through inhibition hypothesis, where slower oscillations are considered to be associated with pulses of inhibition and as such can support inter-areal communication, through phase synchronization and release of inhibition [64,65]. The unified framework based on nested oscillations posits that the communication between two regions is established by phase synchronization of oscillations at lower frequencies (<25 Hz), which serve as a temporal reference frame for information carried by high-frequency activity (>40 Hz) [37,46,66]. Our results speak in favor of this unified framework, demonstrating that top-down predictions from FEF are communicated through oscillatory coherence to the posterior cortex, where they dictate sensory information flow through the nesting of gamma oscillations. This would also be compatible with a predictive routing account, where feedback signaling is thought to carry expectations about sensory stimuli through alpha/beta rhythms, while feedforward signaling through gamma activity becomes amplified during unpredicted and surprising inputs [67].

It is worth mentioning that the original proposals concerning the mediators of inter-areal communication [63] have recently been reappraised [68]. These authors point out that coherence could be an outcome of communication rather than reflect communication and information processing per se [68]. Our findings do seem to be compatible with the suggested revised framework along two dimensions. First, although the framework proposes that coherence is not a physiologically plausible mechanism for mediating communication, it nevertheless accommodates phase shifts as a means of communication. It achieves this by proposing that phase shifts could reliably achieve selective communication at relatively

short timescales, which is in line with our results on phase alignment. Second, the communication-through-resonance theory proposed by the author [68] suggests that neural communication might rely on resonance such that communication is facilitated when a sending population shifts its energy toward the receiver's population intrinsic resonance peak [68]. Accordingly, resonance-driven rhythms that are engaged in feedback processing will involve different timescales and various oscillatory mechanisms requiring non-linear integration processes, which could be carried by cross-frequency couplings as evidence here.

In summary, by emulating attention signaling through FEF-TMS and manipulating bottom-up processing in human participants, this study reveals PAC as a causal mechanism of interactions between low-frequency top-down and high-frequency feedforward signals. Overall, our results corroborate the beta-gamma interplay as an important fingerprint of a FEF-related visual/attention sampling mechanism, in line with recent models and in addition to other, well-documented period sampling mechanisms at alpha and theta-frequency.

## Materials and methods

### Participants

A total of 30 participants took part in the study (18 females; mean age ± SE = 24.6 ± 4.11 years), determined by an a priori power analysis [69], with assumed medium effect size, the desired power level set at 0.90, and an alpha level of 0.05. Furthermore, we also run a post hoc analysis for the used analysis approach, based on power analysis using data simulation for cluster-based permutation tests [70], again confirming the adequacy of our sample size. All participants had normal or corrected-to-normal vision and reported no contraindication to TMS [71] or any neurological, psychiatric, or relevant medical condition. All protocols were performed in accordance with the principles expressed in the Declaration of Helsinki and ethical TMS standards. The study was approved by the Ethical Committee of the College of Science and Engineering (University of Glasgow, 300210149). All participants gave written informed consent to participate in the study.

### Stimuli, task, and procedure

**Main experimental task.** Participants were comfortably seated at a viewing distance of 57 cm from an LCD monitor (144 Hz refresh rate), in a dimly illuminated room, with their chin on a chinrest to ensure a stable head position. The main experimental session consisted of two TMS conditions during which EEG was continuously recorded: Single pulse TMS was applied to either the right FEF or a control area (right M1-foot area), while participants were presented with a moving visual grating (half of the trials, GRATING+) or no grating (other half of the trials, GRATING−). Data was collected in four separate blocks (two blocks per stimulation site, pseudorandomized so that two consecutive blocks would not have the same stimulation site). To uphold the alertness of the participant, a visual task was implemented (see below).

Visual stimuli and tasks were created in Matlab (Psychtoolbox-3). In both TMS conditions (right FEF, right foot M1), the task structure, number of pulses, and pulse timing were kept constant. Specifically, each trial began with a black fixation cross, which, after 1500 ms, turned red to signal that the grating would shortly appear (see Fig 1 for trial structure). After 500 ms, either a continuous circular sine grating contracting inward was presented for 5 s in the central visual field (GRATING+ condition), or no stimulus appeared (GRATING− condition). The task of the participant was to attend to the moving grating when present and to press a button whenever they noticed a glitch in the movement of the grating, while always keeping their eyes on the fixation cross. Motion glitches were implemented to ensure participants remained in an alert state and occurred in 1/3–1/2 of all GRATING+ trials (GLITCH+ condition), while in the rest of the GRATING+ trials, moving gratings were uninterrupted (GLITCH− condition). The duration of the glitch was individualized for each participant, during a separate titration session (see below), to ensure that the perception of the glitch was around the threshold. A TMS pulse was delivered on one of the two TMS sites (depending on the experimental block) in TMS+ conditions, following a jittered interval ranging from 1 s to 4 s after the grating onset (or at the same time into a trial when no grating was presented). In trials where both the TMS pulse and the glitch were present (TMS+/GLITCH+), the timing of the TMS pulse

and the GLITCH coincided. Finally, there were also trials without a TMS pulse (TMS−) to assess gamma activity during the time window of possible TMS delivery (1–4 s).

In total, the experiment consisted of 640 trials and lasted about 90 min. The design led to a maximum of 80 trials per condition to be used in the EEG analyses. Conditions included in the EEG analyses comprised FEF-TMS, control-TMS, or no TMS (TMS±), with/without a stimulus (GRATING±) but no glitches (all GLITCH+ trials excluded for EEG analysis) (see Fig 6 for the details on trial number per condition).

**Titration session.** The duration of the glitch in the sine-grating movement, i.e., the number of frames during which the grating kept still, was individually thresholded for each participant via a blocked staircase procedure. Specifically, the titration session began with a glitch duration value of 20 frames (at 144 Hz, 138.8 ms) in an initial block (N = 10 trials), half of them being target and half of them catch trials (sine grating without the glitch). Subsequently, the percentage of correct trials was calculated, and the glitch duration was adjusted accordingly (100%-96% correct: glitch duration reduced by six frames; 95%–86% correct: glitch duration reduced by four frames; 85%–76% correct: glitch duration reduced by two

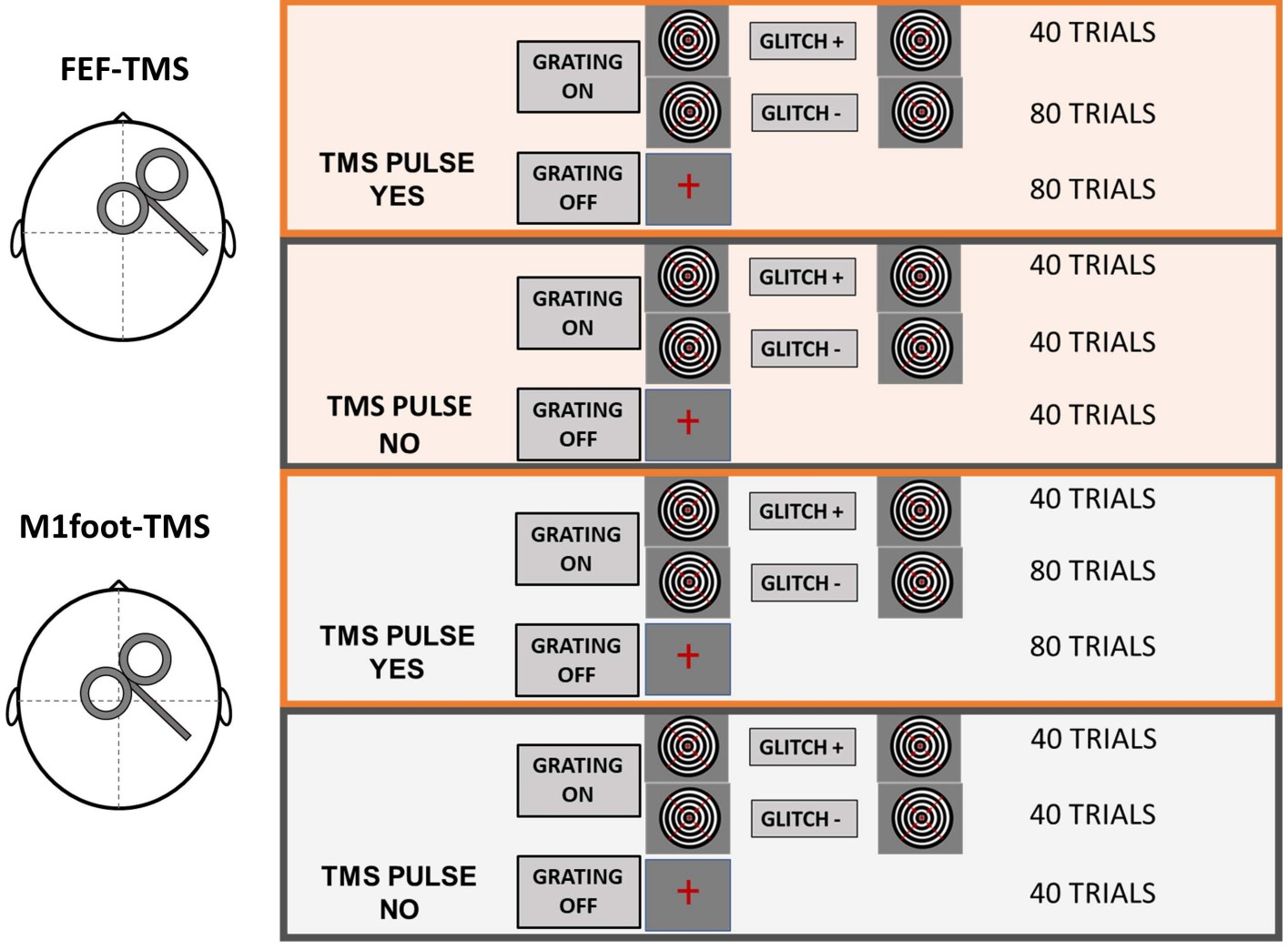

**Fig 6. Experimental design.** Task conditions with respective number of trials.

frames; 75%–56% correct: same glitch duration; 55%–46% correct: glitch duration increased by two frames; 45%–36% correct: glitch duration increased by four frames; 35%–26% correct: glitch duration increased by six frames; 25%–6% correct: glitch duration increased by eight frames; 5%–0% correct: glitch duration increased by 10 frames). The staircase procedure was run for 16 blocks in total (*N* total = 160 trials), and glitch duration values for each block were fitted to a sigmoid function, to identify duration values corresponding to 70% of accurate glitch detection for each participant (mean glitch duration [SD] = 7.14 [2.12] frames), subsequently used in the experiment.

**TMS.** TMS was applied by means of a high-power Magstim Rapid2 machine (Magstim Company, Whitland, UK), with the stimulator connected to a 70 mm standard figure-of-eight coil, triggered remotely using Matlab. TMS intensity was kept constant for the two TMS sites at 65% maximal stimulator output (MSO), based on prior evidence that TMS at this intensity effectively activates FEF [17,36,72]. Likewise, the coil position was kept constant, placed tangentially with the handle of the coil pointing backward and laterally approximately 45 degrees to the interhemispheric line (as in [36]). We would like to point out that although we opted for a fixed TMS intensity that has been shown to be effective in previous FEF-TMS research (see above), it is conceivable that we would have found effects of FEF-TMS on visual activity also with lower TMS intensities. This is because research exploring dose-dependent network effects of FEF-TMS [12] reported that FEF-TMS at 40%–55% MSO can spread to visual areas or even outweigh effects at higher intensities, depending on the visual area under investigation (calcarine versus occipital pole) and presence or absence of visual input. Accordingly, it would be of interest to investigate in future studies dose-dependency of FEF-TMS on oscillatory activity.

To locate the target area, T1-weighted structural magnetic resonance images were acquired via a 3T Siemens Trio Tim scanner (Siemens, Erlangen, Germany) per participant. Subsequently, the stimulation site for the right FEF was individually localized applying the cortex-based alignment (CBA) approach in Brain-Voyager QX 2.8 (Brain Innovation, Maastricht, the Netherlands), described in Ref. [36]. Briefly, anatomical data was used to reconstruct the cortical surface of each participant, which was then aligned to an atlas brain that includes the probabilistic group maps of FEFs. These group maps were then back-transformed to the individual brain anatomy, with the TMS target defined as the center of gravity for the area with the highest probability [73,74]. FEF coordinates were on average (± SD) x = 31.2 ± 3.7, y = −9.7 ± 2.9, z = 49.74 ± 5.1. The control area (M1-foot area) was localized using averaged Talaraich coordinates identified in the literature (x = 6.8, y = −14.8, z = 69.8) [75]. Finally, to target the right FEF and right M1-foot area, and to keep coil position and orientation constant, Talairach coordinates obtained with the CBA (and average coordinates for the control area) were imported to a frameless stereotactic neuro-navigation system (Brainsight; Rogue solution).

### TMS-EEG recordings—acquisition and pre-processing

EEG activity was recorded using a TMS-compatible EEG system (BrainAmp MRplus, BrainProducts), acquiring EEG from 61 TMS-compatible Ag/AgCl multitrode electrodes (EasyCap GmbH, Herrsching, Germany) mounted on a cap and positioned according to the 10–10 International System. Additional electrodes were used as ground (TP9) and recording reference (AFz). The signal was acquired at a sampling rate of 5,000 Hz, and bandpass filtered at 0.1–1,000 Hz. Electrode impedance was maintained below 5 KΩ. An additional electrode was positioned on the outer canthus of the left eye to record eye movements.

TMS artifacts were identified and removed using an open-source EEGLab extension, the TMS-EEG signal analyzer [76]. First, EEG data were epoched around sine-grating stimulus appearance or TMS pulse onset (between −2,000 ms and 1,000 ms). In trials where TMS pulse was not present, an additional marker in the TMS-timing window was added to allow for epoching. Afterward, a baseline correction was applied by subtracting the average of the entire epoch from each data point (demeaning the data). Data around the TMS pulse (−1 ms +10 ms) was removed and replaced with zeros and cubic interpolation of the removed window was performed before down-sampling the data (from 5,000 Hz to 1,000 Hz). At this point, data were visually inspected to remove noisy trials. Interpolated data around the pulse was again removed before individual component analysis (ICA). Specifically, a fast ICA algorithm was used (pop_tesa_fastica function) to

identify individual components representing artifacts, along with automatic component classification (pop_tesa_compselect function), where each component was subsequently manually checked and reclassified when necessary. In this first round of ICA, only components with large amplitude artifacts, such as TMS-evoked scalp muscle artifacts, were eliminated ($M_{removedIC}$ ± SD = 5.95 ± 1.83). Data around the TMS pulse were again interpolated (cubic interpolation) before applying pass-band (between 1 Hz and 100 Hz) and stop-band (between 48 Hz and 52 Hz) Butterworth filters. Subsequently, interpolated data were again removed before the second round of ICA, to remove all other artifacts, such as blinks, eye movement, persistent muscle activity, and electrode noise ($M_{removedIC}$ ± SD = 18.36 ± 5.95). Then, the TMS pulse period was again interpolated (cubic interpolation), and data was re-referenced to the average of all electrodes. Finally, single trials were visually inspected, and those containing residual TMS artifacts were removed. The described TMS artifact removal procedure was applied to all EEG data. On average, less than 15% of all epochs were removed (M = 14.19%, SD = 5.85%). Moreover, the resulting trial number across different stimulation sites (FEF versus control site) was not statistically different for both the grating-present (Mfef = 69.6 SE =1.54; Mctrl = 71.4 SE = 0.86, $t$ = 1.228, $p$ = 0.229) and grating-absent condition (Mfef = 62.8 SE = 1.72; Mctrl = 62.9 SE = 1.68, $t$ = 0.057, $p$ = 0.955).

### Time-frequency analysis: Effects of visual stimuli (GRATING±) on oscillatory amplitude

As a first step, to confirm that the continuous sine grating did indeed lead to an increase of gamma activity, a time-frequency analysis was performed on the signal epoched around the grating stimulus onset. Specifically, complex Morlet wavelet convolution was applied to the signal, with Morlet wavelet peak frequencies ranging from 3 Hz to 80 Hz in 60 logarithmic steps. The full width at half-maximum ranged from 500 ms to 200 ms with increasing wavelet peak frequency. Subsequently, power values were normalized by decibel conversion, with the baseline period from −500 ms to −200 ms before grating stimulus onset. Finally, time-frequency differences in power between the trials with and without the sine-grating stimulus in parieto-occipital cluster (cluster electrodes: O2, PO4, PO8, Oz, POz, PO3, PO7, O1) were compared via cluster-based permutation testing. Specifically, z-scores for each individual data point in the time-fequency (TF) plane were obtained by comparing the time-frequency map differences (trials with versus without the sine grating) with permuted ($N$ = 1,000) random differences (obtained by randomizing the sign of difference map for every subject). Differences with z-values corresponding to $p < 0.05$ were retained as significant. Subsequently, the results were cluster-corrected, where the cluster threshold was determined by the 5% biggest cluster size in the null distribution of permuted z-scores (thus $p < 0.05$).

### Phase realignment: Effects of FEF-TMS (versus control-TMS) on occipito-parietal activity

Next, we aimed to replicate the findings of Veniero and colleagues [36] of enhanced beta phase realignment in connected posterior areas after FEF activation by a TMS pulse. We used ITPC (as in [36]) to track the influence of FEF stimulation on the posterior cortex across time-frequency space. ITPC measures the extent to which phase angles at each time-frequency-electrode point across the trials are nonuniformly distributed in polar space [77]. Phase angles were obtained by the same wavelet convolution procedure used for the time-frequency power analysis. Likewise, the same statistical permutation-based cluster analysis used for comparing amplitude differences was used for ITPC-difference plots between the two TMS protocols (FEF-TMS, M1-foot-TMS), separately for the stimulus present (GRATING+) and stimulus absent (GRATING−) conditions. The same posterior electrodes as for the time-frequency amplitude analysis were used, but separately for the right (O2, PO4, PO8, Oz, POz) and left electrode cluster (Oz, POz, PO3, PO7, O1), given the right-lateralized TMS over FEF/foot areas.

### Connectivity changes: Effects of FEF-TMS (versus control-TMS) on fronto-posterior connectivity

Inter-areal connectivity was estimated in the sensor space between frontal and parieto-occipital electrodes of both hemispheres (extending the analyses of [36]) via the wPLI [78]. This is a measure of phase lag-based connectivity, which accounts for non-zero phase lag/lead relations between two signal time series. It defines connectivity as the absolute value

of the average sign of phase-angle differences and additionally deweights vectors that are closer to the real axis such that those vectors have a smaller influence on the final connectivity estimate. By extension, this measure is insensitive to volume conduction and noise of a different origin, considered optimal for exploratory analysis as it minimizes type-I errors [79]. To obtain wPLI values, time series data was first transformed into the time-frequency domain via convolution with a family of complex Morlet wavelets (the number of cycles increased from five to 18 in logarithmic steps). Therefore, for frequencies ranging from 3 Hz to 30 Hz in 1 Hz steps, first convolution by frequency-domain multiplication was performed and then the inverse Fourier transformation was taken. Phase was defined as the angle relative to the positive real axis, and phase differences were then computed between all possible pairs of electrodes. Finally, wPLI was calculated as the absolute value of the average sign of phase-angle differences, whereby vectors closer to the real axis were de-weighted. Once wPLI values were extracted for every frequency bin (3–30 Hz in 1 Hz steps) and epoch, they were averaged to obtain distinct values for each experimental condition of interest (FEF and M1-foot-TMS when sine grating was present or absent) in the higher alpha/lower beta frequency band identified as significant via ITPC (13–20 Hz, see results), and for the 300 ms after the stimulation. This time window was informed by the duration of significant effects in ITPC and allows at the same time for at least four entire cycles of the high alpha/low beta activity in the assessment of connectivity. Subsequently, non-parametric permutation-based analysis (1,000 iterations) was performed to compare the connectivity differences between the two TMS protocols and to obtain phase connectivity difference maps of distinct electrode pairs. For these pairwise calculations of fronto-posterior connectivity, we included the following electrodes of the right (stimulated) hemisphere (frontal: AF4, AF8, F2, F4, F6, F8, FC2, FC4, FC6; posterior: Pz POz Oz P2 P4 P6 P8 PO4 PO8 O2), as well as homologous electrodes of the left hemisphere. Sensor differences with z-values corresponding to $p < 0.05$ were retained as significant. The connectivity index of each condition was then estimated using the formula: CI = sig_pos/sp_total, where sig_pos are connections that are significantly higher after FEF-TMS with respect to M1foot-TMS, and sp_tot are all possible connections [80,81]. Furthermore, another permutation test was introduced to calculate the significance threshold for confidence interval (CI). Specifically, wPLI matrices of all the electrode pairs and experimental conditions were randomly permuted and compared 1,000 times to obtain the distribution of randomly obtained differences in wPLI. Connectivity indices that exceeded a 95% CI were considered statistically significant (connectivity threshold = 0.05).

## Phase-amplitude coupling

Finally, the hypothesized interaction between the top-down signal emulated by the TMS pulse and feedforward sensory input was measured via PAC, with the idea that the phase of low-frequency (alpha/beta) rhythms should modulate the amplitude of high-frequency (gamma) oscillations. PAC was quantified via the MI [82]. Briefly, the signal was first filtered at the two frequency ranges under analysis (low frequencies = 5–20 Hz with 4 Hz bandwidth; high frequencies = 30–80 Hz with 10 Hz bandwidth). Then, the Hilbert transform was applied to obtain the phase time series for the lower frequency range and the time series of the amplitude envelope for higher frequencies. Next, these phases were binned, and the mean amplitude over each bin was calculated and normalized by the sum over all the bins, resulting in mean amplitude distribution over phase bins. Finally, the MI was obtained by measuring the divergence of the observed amplitude distribution from the uniform distribution. MI was calculated for every cross-frequency point in time window of interest (0–300 ms post-TMS), concatenated between trials, for the right parieto-occipital cluster (POZ, OZ, O2, PO4, PO8), and identified as significant with amplitude and ITPC analyses. The same statistical permutation-based cluster analysis used for the amplitude and ITPC analyses was used for PAC-difference maps between two TMS sites, separately for grating-present and grating-absent conditions.

## Sanity checks and control analyses

To rule out several caveats associated with phase-based analyses as used in the present work, we run a number of control analyses and sanity checks, all supportive of our conclusions. These are reported in detail in the Supporting information, see S2-S5 Figs.

 PLOS Biology

## Supporting information

**S1 Fig. Time-frequency analysis for the whole grating duration: Grating effects. (A)** Left panel: Raw differences in time-frequency plots of the posterior electrodes (O2, O1, POz, Oz, PO8, PO7, PO4, PO3) between grating-present and grating-absent condition (GRATING±). Frequency range for the analysis (y-axis) is from 5–80 Hz. Time range for the analysis (x-axis) is from −200–5,000 ms, where 0 is the time point of the grating onset. Right panel: Z-scores of the permutation-based analysis between grating-present and grating-absent condition. Significant clusters are framed with the black line. **(B)** Topographies of the significant clusters of the amplitude differences in the gamma frequency band (lower). diff = difference; dB = decibel; Hz = hertz; *t* = time.
(DOCX)

**S2 Fig. Phase-angle differences.** Potential differences in the preferred phase lag angle differences between the electrode sites are not responsible for the obtained connectivity differences. Specifically, we did not find significant differences between the electrodes and frequencies where we found significant connectivity changes via wPLI in the grating-present (all ts < 1.702; all ps > 0.090) and grating-absent condition (all ts < 1.287; all ps > 0.203). **(A)** Grating-absent condition. Phase-angle differences for the FEF-TMS (left) and M1-TMS condition (right) in polar coordinates between one of the significant electrode pairs (AF8-PO4) at 16 Hz. Each line represents phase-angle differences of a single participant. **(B)** Grating-present condition. Phase-angle differences for the FEF-TMS (left) and M1-TMS condition (right) in polar coordinates between one of the significant electrode pairs (AF8-PO4) at 16 Hz. Each line represents phase-angle differences of a single participant.
(DOCX)

**S3 Fig. Power-frequency spectra** showing the presence of high alpha-low beta band activity which is a prerequisite for phase-analyses in these frequency bands. **(A)** Grating-absent condition. Power-frequency spectrum for the FEF-TMS (left) and M1-TMS condition (right) averaged across participants. Gray square represents the frequency window of interest. **(B)** Grating-present condition. Power-frequency spectrum for the FEF-TMS (left) and M1-TMS condition (right) averaged across participants. Gray square represents the frequency window of interest.
(DOCX)

**S4 Fig. Time-frequency power and ITPC analysis for the no-TMS trials across different blocks: Control analysis.** No evidence for between-block differences in power or ITPC (only no-TMS trials included) ruling out potential confounding effects of block on the TMS results. **(A)** Left panel: Raw differences in time-frequency plots of the posterior electrodes (O2, O1, POz, Oz, PO8, PO7, PO4, PO3) between different blocks of the grating-absent condition (GRATING−). Frequency range for the analysis (y-axis) is from 5–80 Hz. Time range for the analysis (x-axis) is from −200–1,000 ms, where 0 is the time point of the grating onset. Right panel: Z-scores of the permutation-based analysis between different block of the grating-absent condition. No significant clusters were identified. **(B)** Left panel: Raw differences in time-frequency plots of the posterior electrodes (O2, O1, POz, Oz, PO8, PO7, PO4, PO3) between different blocks of the grating-present condition (GRATING+). Frequency range for the analysis (y-axis) is from 5–80 Hz. Time range for the analysis (x-axis) is from −200–1,000 ms, where 0 is the time point of the grating onset. Right panel: Z-scores of the permutation-based analysis between different blocks of the grating-present condition. No significant clusters were identified. **(C)** Left panel: Inter-trial phase coherence (ITPC) differences of the posterior cluster in the stimulated (right) hemisphere (electrodes: O2, POz, Oz, PO8, PO4) between different blocks of the grating-absent condition (GRATING−). The time range for the analysis (x-axis) is from −200–1,000 ms, where 0 point is the timing of the TMS pulse. The frequency (y-axis) is from 5–80 Hz. Right panel: Z-scores of the permutation-based statistical analysis between different blocks of the grating-absent condition (GRATING−). No significant clusters were identified. **(D)** Left panel: ITPC differences of the posterior cluster in the stimulated (right) hemisphere (electrodes: O2, POz, Oz, PO8, PO4) between different blocks of the grating-present condition

(GRATING+). The time range for the analysis (x-axis) is from −200–1,000 ms, where 0 point is the timing of the TMS pulse. The frequency (y-axis) is from 5–80 Hz. Right panel: Z-scores of the permutation-based statistical analysis between different blocks of the grating-present condition (GRATING+). No significant clusters were identified. diff = difference; dB = decibel; Hz = hertz; $t$ = time.
(DOCX)

**S5 Fig. Weighted phase lag index (wPLI) and phase-amplitude coupling (PAC) for the no-TMS trials across different blocks: Control analysis. No evidence for between-block differences in wPLI or PAC (only no-TMS trials included) ruling out potential confounding effects of block on the TMS results. (A)** Left: Grating-absent Condition. Connectivity matrix measured by the wPL) across all electrodes included in the regions of interest for the grating-absent condition in the higher alpha/lower beta range (13–20 Hz), in the right (stimulated) hemisphere. No significant differences in interregional coupling across different experimental blocks were identified. Right: Grating-present Condition. Connectivity matrix measured by the wPLI across all electrodes included in the regions of interest for the grating-present condition in the higher alpha/lower beta range (13–20 Hz), in the right (stimulated) hemisphere. Red/blue ink indicates significant differences in interregional coupling across different experimental blocks. **(B)** Grating-absent condition. Left panel: Raw differences in modulation index (MI) plots of the posterior cluster in the stimulated (right) hemisphere (electrodes: O2, POz, Oz, PO8, PO4) between different block of the no-TMS trials. The frequency for lower phase frequency is shown in the y-axis. The frequency for the higher gamma amplitude- in the x-axis. Right panel: Z-scores of the permutation-based analysis between different experimental blocks. No significant clusters were identified. **(C)** Grating-present condition. Left panel: Raw differences in MI plots of the posterior cluster in the stimulated (right) hemisphere (electrodes: O2, POz, Oz, PO8, PO4) between different block of the no-TMS trials. The frequency for lower phase frequency is shown in the y-axis. The frequency for the higher gamma amplitude in the x-axis. Right panel: Z-scores of the permutation-based analysis between different experimental blocks. No significant clusters were identified.
(DOCX)

## Author contributions

**Conceptualization:** Jelena Trajkovic, Satu Palva, Vincenzo Romei, Gregor Thut.

**Investigation:** Jelena Trajkovic.

**Methodology:** Jelena Trajkovic, Domenica Veniero, Simon Hanslmayr, Satu Palva, Gabriela Cruz, Vincenzo Romei, Gregor Thut.

**Supervision:** Gregor Thut.

**Writing – original draft:** Jelena Trajkovic, Domenica Veniero, Simon Hanslmayr, Satu Palva, Gabriela Cruz, Vincenzo Romei, Gregor Thut.

**Writing – review & editing:** Jelena Trajkovic, Domenica Veniero, Simon Hanslmayr, Satu Palva, Gabriela Cruz, Vincenzo Romei, Gregor Thut.

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
