## [Editor Report · Decision Letter 0]

21 May 2024

Dear Dr Trajkovic, 

Thank you for submitting your manuscript entitled "Top-down and bottom-up interactions rely on nested brain oscillations." for consideration as a Research Article by PLOS Biology.

Your manuscript has now been evaluated by the PLOS Biology editorial staff as well as by an academic editor with relevant expertise and I am writing to let you know that we would like to send your submission out for external peer review.

Once your full submission is complete, your paper will undergo a series of checks in preparation for peer review. After your manuscript has passed the checks it will be sent out for review. To provide the metadata for your submission, please Login to Editorial Manager (https://www.editorialmanager.com/pbiology) within two working days, i.e. by May 23 2024 11:59PM.

Kind regards,

Christian

Christian Schnell, PhD

Senior Editor

PLOS Biology

cschnell@plos.org

---

## [Decision Letter · Decision Letter 1]

23 Aug 2024

Dear Dr Trajkovic,

Thank you for your patience while your manuscript "Top-down and bottom-up interactions rely on nested brain oscillations." was peer-reviewed at PLOS Biology. Please allow me to apologize for the long delay in getting back you tou. It was more challenging than usual to find reviewers and then those reviewers also needed more time during the summer period. Your manuscript has now been evaluated by the PLOS Biology editors, an Academic Editor with relevant expertise, and by several independent reviewers. 

In light of the reviews, which you will find at the end of this email, we would like to invite you to revise the work to thoroughly address the reviewers' reports.

As you will see below, the reviewers are overall positive, but they raise a number of concerns regarding the lack of methodological details, additional required analyses, and the issue that you do not report behavioral effects. 

Given the extent of revision needed, we cannot make a decision about publication until we have seen the revised manuscript and your response to the reviewers' comments. Your revised manuscript is likely to be sent for further evaluation by all or a subset of the reviewers.

**IMPORTANT - SUBMITTING YOUR REVISION**

*Re-submission Checklist*

*Published Peer Review*

*PLOS Data Policy*

*Blot and Gel Data Policy*

Sincerely,

Christian

Christian Schnell, PhD

Senior Editor

PLOS Biology

cschnell@plos.org

REVIEWS:

Reviewer #1: The article entitled "Top-down and bottom-up interactions rely on nested brain oscillations" by Trajkovic et al. provides supporting evidence for frequency-specific top-down modulation of bottom-up sensory signals in human participants. 

Using single pulse transcranial magnetic stimulation (TMS) of the Frontal Eye Field (FEF), they show increased phase realignment of activity at alpha/beta frequency for ipsilateral occipito-parietal sensors and increased alpha/beta phase-synchronization between frontal and occipito-parietal sensors. These effects are significant with or without visual stimulation but phase-amplitude coupling between frontal and occipital oscillatory activity is only significant during visual stimulation and shows that occipital gamma frequency power is aligned to alpha/beta phase.

This article seems at first like an implementation of their previous study with Joachim Gross (Veneiro et al., NatCommun2022) reporting that TMS activation of FEF induces beta phase-reset over occipital sites, although it brings important information that could not be analyzed from the 2022 study. 

The advance resides in the fact that they causally tested previous non-human primate observations of cross-frequency coupling between top-down beta and bottom-up gamma frequency oscillations (e.g. Richter et al., 2017). 

They used TMS in the presence versus absence of sensory activity and explore frequency interactions up to 80Hz thus allowing them to look for effect of top-down alpha/beta on bottom-up gamma. However, the functional connectivity measure used (wPLI) is not directional.

The paper reads well and results are clearly reported, although some aspects of the results shall be discussed further. 

In addition, results on TMS effects on behavioral performance shall also be reported and discussed. 

The points to address are detailed below:

1. PAC of alpha/beta to gamma oscillations is associated to a mechanism through which top-down/feedback modulates bottom-up/feedforward signals for optimal performance. However, results on behavioral performance are not presented at all. For the interest of the reader, I strongly recommend the authors to report the behavioral effect of TMS when grating present (either on performance or on reaction time by contrast with M1foot-TMS and/or without TMS) and include discussion about these and comparison with literature. As FEF-TMS is reported to enhance alpha/beta phase synchronization and occipital gamma power, one clearly expect behavior to follow e.g. Rohenkohl et al. Neuron 2018; Parto-Dezfouli et al., Cell Rep. 2023. 

2. On Figure 4B and C, it is clear that FEF-TMS effects on alpha/beta phase-synchronization between frontal and parieto-occipital sensors differ. In Grating+ condition, only AF4-8 show increased synchronization with almost all parieto-occipital sensors, while in Grating- condition, more than half of frontal sensors show increased synchronization with PO4-8-2 only. These topographic differences shall be discussed further.

3. The authors refer to prominent theories of oscillatory communication (paragraph lines282-295 p10). It would make the discussion on nested oscillations more appealing to the reader (and potentially increase the impact of the article) when considering more alternative (although not incompatible) views e.g. Vinck et al., Neuron 2023; Bastos et al., PNAS 2020.

Reviewer #2: This is a very strong study, addressing an important question in cognitive neuroscience with very robust methods. I only have couple of very minor points:

1. Sample size - 30 participants were tested. What was this based on? Were power calculations carried out prior to the study.

2. TMS intensity - could this be expanded upon? Is there any reason to expect that low vs high intensities would have different effects, as has been shown with single pulse TMS?

Reviewer #3: In "Top-down and bottom-up interactions rely on nested brain oscillations", Trajkovic and colleagues build on considerable human and NHP work demonstrating that top-down influences arise via alpha/beta synchrony while bottom-up information is carried via gamma synchrony and ask how these two signals are related. They specifically examine whether these two signals are coordinated via increased coupling between alpha/beta phase and gamma amplitude. The results are relatively straightforward and make a theoretical advance into our understanding of top-down influences on perception. I have a few concerns that the authors should address, which I detail below.

(1) One potential concern about comparing phase-based connectivity measures (here, wPLI) that calculate the magnitude of the phase lag across conditions is that any differences between conditions could be due to a difference in the preferred phase angle difference between electrode sites, which is not necessarily indicative of increased connectivity. For more details, please refer to chapter 26 (~page 350 in the hardcover edition) of ref [65]. It would be prudent for the authors to rule out this possibility by checking the phase lag angle difference in both conditions and ensuring that they are not significantly different.

(2) There are a few additional important details about the methods and analyses that are missing. First, the time windows and frequency ranges reported vary across different analyses with little to no justification. The effects of TMS on alpha/beta band ITPC lasted for 250 or 400ms, depending on condition, but connectivity was calculated over a 300ms post-TMS window, and PAC was tested in a 250ms post-TMS window in a wider frequency range than was reported earlier in the manuscript. There is also an incongruity between the gamma range in which PAC was tested (30-80Hz) and the gamma range in which significant results were observed (60-90Hz). Second, a substantial number of trials are from the no-TMS condition, which occurred in both the FEF-TMS and M1-TMS blocks. I could not find any results from these trials. It would be instructive to confirm that the gamma power, alpha/beta ITPC, and alpha/beta-gamma PAC results did not differ between the no-TMS trials in the two blocks to rule out a general effect of the blocks that is not specific to the TMS pulse. Third, the authors should provide a justification for the brief time window used to calculate PAC, as, depending on the method, some authors advise using a window of up to 10s.

(3) There are a few caveats to the ITPC analysis that the authors need to consider. First, phase measures are only valid when power is sufficient. It would be a good sanity check to see the power at the frequencies of interest, perhaps as a supplementary figure. Second, related to point (2) above, it is not clear how many trials are in each of the conditions of interest. ITPC is sensitive to unbalanced trial numbers, so the authors should ensure that the number of trials is comparable between M1 and FEF conditions and report this information. 

(4) Presumably the pattern of results described here is exciting because ultimately it is adaptive for behavior; yet, there is no discussion of how TMS, which was intended to emulate top-down signals, affected behavior. The authors have a behavioral measure that might be useful in glitch detection performance. They even carefully titrate behavior prior to the experiment to ensure that participants correctly detect the glitch 70% of the time. Wouldn't one expect increased performance in the FEF TMS condition? How should our interpretation of the results change if this is not the case?

(5) Some minor points:

* There is some relevant work by Riddle and colleagues (Riddle et al., 2019, JoCN; Riddle et al., 2020, Current Biology) that the authors should consider including in their introduction/discussion.

* The sentence starting on p. 6, line 181 is confusing and potentially misleading; one can take it to mean that gamma-band activity over posterior sites is related to phase realignment.

* Figure 1 - I believe "ANS" should be "AND"

* The citation format in parts of the Methods section is inconsistent with the rest of the manuscript.

---

## [Editor Report · Decision Letter 2]

24 Jan 2025

Dear Jelena,

Thank you for your patience while we considered your revised manuscript "Top-down and bottom-up interactions rely on nested brain oscillations." for publication as a Research Article at PLOS Biology. This revised version of your manuscript has been evaluated by the PLOS Biology editors, the Academic Editor [and the original reviewers -EDIT AS APPLICABLE].

Based on our Academic Editor's assessment of your revision, we are likely to accept this manuscript for publication, provided you satisfactorily address the remaining point raised by the Academic Editor. Please also make sure to address the following data and other policy-related requests:

* We would like to suggest a different title to improve its accessibility for our broad audience: "Top-down and bottom-up interactions rely on nested brain oscillations to mediate rhythmic visual attention sampling"

* Please add the links to the funding agencies in the Financial Disclosure statement in the manuscript details.

* Please include information in the Methods section whether the study has been conducted according to the principles expressed in the Declaration of Helsinki.

* Simon Hanslmayr needs to declare in the "Competing Interests" section that he is an Editorial Board Member of PLOS Biology.

* Please note that per journal policy, the model system/species studied should be clearly stated in the abstract of your manuscript, for example "in human the visual system".

* DATA POLICY:

Regardless of the method selected, please ensure that you provide the individual numerical values that underlie the summary data displayed in the following figure panels as they are essential for readers to assess your analysis and to reproduce it.

* CODE POLICY

We expect to receive your revised manuscript within two weeks. 

*Published Peer Review History*

*Press*

Sincerely,

Christian

Christian Schnell, PhD

Senior Editor

cschnell@plos.org

PLOS Biology

Academic Editor remarks:

The response to the reviewers has improved the manuscript particularly with respect to the recent publication from the Vinck lab (ref 69). However, in the discussion the modified text is not very clear to a non-specialist and I recommend the following modifications:

"It is worth mentioning that the original proposals concerning the mediators of inter-areal communication (REF Fries et al), have recently been reappraised (REF 69). These authors point out that coherence could be an outcome of communication rather than reflect communication and information processing per se. Our findings do seem to be compatible with the suggested revised framework along two dimensions. First, although the framework proposes that coherence is not a physiologically plausible mechanism for mediating communication, it nevertheless accommodates phase shifts as a means of communication. It achieves this by proposing that phase shifts could reliably achieve selective communication at relatively short timescales, which is in line with our results on phase-alignment. Second, the communication-through-resonance theory proposed by (REF 69) suggests that neural communication might rely on resonance such that communication is facilitated when a sending population shifts its energy toward the receiver’s population intrinsic resonance peak (69). Accordingly, resonance-driven rhythms that are engaged in feedback processing will involve different timescales and various oscillatory mechanisms requiring non-linear integration processes, which could be carried by cross-frequency couplings as evidence here”.

---

## [Editor Report · Decision Letter 3]

12 Feb 2025

Dear Jelena,

Thank you for the submission of your revised Research Article "Top-down and bottom-up interactions rely on nested brain oscillations to shape rhythmic visual attention sampling." for publication in PLOS Biology. On behalf of my colleagues and the Academic Editor, Henry Kennedy, I am pleased to say that we can in principle accept your manuscript for publication, provided you address any remaining formatting and reporting issues. These will be detailed in an email you should receive within 2-3 business days from our colleagues in the journal operations team; no action is required from you until then. Please note that we will not be able to formally accept your manuscript and schedule it for publication until you have completed any requested changes.

PRESS

Sincerely, 

Christian

Christian Schnell, PhD

Senior Editor

PLOS Biology

cschnell@plos.org